# Ice-marginal proglacial lakes enhance outlet glacier velocities across Greenland
Connie M. Harpur ✉, Mark W. Smith, Jonathan L. Carrivick, Duncan J. Quincey & Liam Taylor

Ice-marginal lakes can alter glacier dynamics, typically accelerating mass loss. The number and size of lakes bordering the Greenland Ice Sheet have increased over recent decades, but their influence on the velocity of outlet glaciers remains largely unquantified. Here, we compare the longitudinal velocity profiles of 102 lake- and land-terminating glaciers across the Greenland Ice Sheet. We find that lake-terminating glaciers are 231% faster at the terminus and exhibit enhanced ice speeds up to ~3.5 km inland. Furthermore, ~44% of lake-terminating glaciers accelerate towards their termini, compared to only ~4% of land-terminating glaciers. The relationship between lake size and glacier behaviour is complex, but glaciers terminating in the largest lakes generally exhibit greater rates of down-ice acceleration than those terminating in the smallest lakes. Together, these results demonstrate that outlet glaciers respond dynamically to lakes at their termini, which should be accounted for in models of ice sheet evolution.

Ice loss from the Greenland Ice Sheet (GrIS) has been increasing since the 1980s and now constitutes a primary source of global sea level rise[1,2]. More than half of this mass loss occurs via ice sheet outlet glaciers, which are accelerating, thinning and retreating in response to climate and ocean warming[3–6]. These changes have significantly reconfigured the ice sheet margin, particularly with a growth in the number and size of ice-marginal proglacial lakes (IMLs), which form as meltwater accumulates in the numerous topographic overdeepenings revealed during ice sheet retreat[7–11]).

IMLs have the propensity to alter glacier dynamics and enhance rates of ice mass loss[12–18]. Specifically, much like in marine settings, IMLs can induce flotation and submarine melting of a glacier terminus, thereby facilitating calving, modifying subglacial water pressures, and reducing buttressing of glacier flow. The subsequent decrease in effective pressure can then propagate up-glacier, leading to sustained ice flow acceleration[19]. These processes have largely been interpreted from mountain glacier observations; in the Himalaya, for example, comparison of neighbouring lake- and land-terminating glaciers has revealed contrasting patterns of velocity, whereby lake-terminating glaciers flow twice as quickly as their land-terminating counterparts and even accelerate towards their termini in a manner typical of tidewater glaciers[14,16]. This type of extensional flow promotes thinning and rapidly advects ice towards a glacier front, where it is lost through processes of frontal ablation[20].

Although similar trends have been widely documented elsewhere (c.f.[21]), the role of IMLs in modulating GrIS outlet glacier dynamics remains relatively unexplored. In southwest Greenland, Mallalieu et al. (2021)[15] recorded progressive retreat of lake-contact areas of the ice margin between 1987 and 2015, at rates four times faster than terrestrial

margins. In the same region, Holt et al. (2024)[22] observed pronounced terminus thinning of a lake-terminating outlet glacier, which led to increased flow velocity up to 15 km inland over the eight-year study period (2013–2021) and induced a sustained pattern of towards-terminus acceleration. Critically, these effects appear to be prevalent across the ice sheet, with ice surface velocities at lake-contact margins, including those tangential to flow, averaging 25% faster than at terrestrial margins[13]. However, the influence of IMLs on specifically outlet glacier velocities across the broader GrIS is unknown, and the inland reach of this effect remains unquantified.

Approximately 10% of the GrIS margin is now bounded by freshwater, and the proportion of the ice sheet draining into lakes, as opposed to marine or terrestrial settings, will likely increase over the coming decades[13]. Despite this, poor constraints on the influence of IMLs on Greenlandic outlet glaciers leave lake effects unaccounted for in models of ice sheet evolution. Since ice dynamics are set to remain a principal driver of GrIS mass loss[23], understanding the role of IMLs is crucial for refining projections of Greenland's future sea level contributions[21].

This study therefore aims to assess the influence of IMLs on outlet glacier flow across the Greenland Ice Sheet. We use pre-existing datasets of ice surface velocity and IML characteristics to produce longitudinal velocity profiles of lake-terminating outlet glaciers within all ice sheet sectors, and quantify the influence of IMLs by way of comparison with an equivalent set of land-terminating glaciers. Further, given the anticipated expansion of new and existing IMLs, we assess the relationship between lake area and glacier flow regime. Our results reveal a 231% increase in the terminus velocity of lake-terminating glaciers compared to land-terminating glaciers

School of Geography, Faculty of Environment, University of Leeds, Leeds, UK. ✉e-mail: gycmh@leeds.ac.uk

during 2017. This enhancement effect diminishes with distance inland but is evident up to ~3.5 km from the ice sheet margin, showing that lake effects propagate beyond the calving front. Additionally, although land-terminating glaciers typically decelerate across their lower reaches, ~44% of the lake-terminating glaciers accelerate longitudinally. Overall, these findings demonstrate that GrIS outlet glaciers respond dynamically to IMLs, albeit variably, with implications for the inclusion of IMLs in ice sheet models.

## Results
### Contrasting behaviour of lake- and land-terminating glaciers
Our analysis reveals contrasting longitudinal velocity profiles at lake-terminating and land-terminating outlet glaciers (Fig. 1a). For each up-glacier distance, we calculated the median velocities for lake- and land-terminating glaciers. Median velocities were significantly higher at lake-terminating glaciers than land-terminating glaciers ($p = 0.04$), with overall median values of 43 m yr$^{-1}$ and 36 m yr$^{-1}$, respectively. Whilst the two groups display insignificant differences at sample boxes between 5000 m and 9500 m up-ice, the aggregated velocity profiles diverge towards the ice sheet margin such that, from 3.5 km inland, lake-terminating glaciers are both significantly ($p \leq 0.01$) and progressively faster than their land-terminating counterparts. These significant differences are consistently observed in every year between 2015 and 2019. In years 2016, 2018 and 2019, lake-terminating glaciers are significantly faster up to 5 km inland (Table 1).

The downstream divergence in profiles shown in Fig. 1a reflects a −61% decrease in median land-terminating glacier velocity, compared to a −23% decrease at lake-terminating glaciers over the lowermost 5 km. As a result, the difference in median velocity between lake- and land-terminating glaciers is greatest within the terminus region, reaching 30 m yr$^{-1}$ at 500 m inland, where lake- and land-terminating glaciers display median velocities of 43 m yr$^{-1}$ and 13 m yr$^{-1}$, respectively. Here, land-terminating glaciers show consistently low velocities with little variability amongst the population (interquartile range, IQR = 19 m yr$^{-1}$); in contrast, lake-terminating glaciers display a wide spread of velocities (IQR = 68 m yr$^{-1}$), indicative of increasingly heterogenous behavior with proximity to the terminus. These patterns are consistent across all six ice sheet regions but are statistically significant only in the southwest (SW), north (NO), and northeast (NE), where sample numbers are > 5 (Fig. S1; Table S1). To test whether our results are specific to the period 2015−2019, or instead persistent, we also analysed velocity during the year 2000 at 500 m up-ice from the termini. Like in 2015−2019, lake-terminating glaciers were significantly ($p < 0.01$) faster at 500 m up-ice (51 m yr$^{-1}$) than land-terminating glaciers (29 m yr$^{-1}$) during 2000 (Fig. S2).

When the behaviour of individual glaciers is assessed, we find substantial differences in the direction and magnitude of near-terminus velocity changes at lake-terminating and land-terminating glaciers, calculated as the percentage change in velocity between the 2000 m and 500 m sample locations (Fig. 1b). Land-terminating glaciers predominantly decelerate towards their termini (median deceleration −50%, maximum deceleration −88%), with only 3.9% ($n = 4$) of the sample group exhibiting down-ice accelerations (median acceleration 17%, maximum acceleration 50%). Contrastingly, lake-terminating glaciers tend to maintain higher velocities or gain speed towards their termini; of the 97 lake-terminating glaciers with data points at both 2000 m and 500 m along the flowline, 56% decelerated

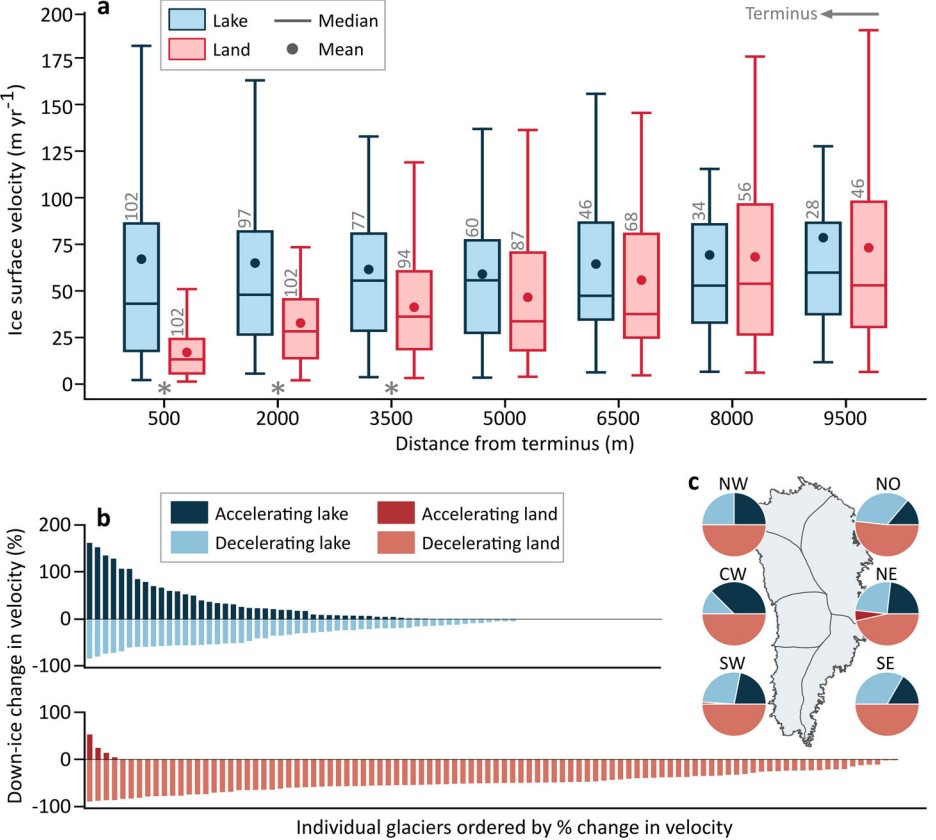

**Fig. 1 | Lake- and land-terminating outlet glacier velocities. a** Median box-sampled velocity at each flowline location, aggregated from all lake-terminating (blue) and land-terminating (red) glaciers. The IQR of velocity values across the sample population is shown by the box height, with capped bars representing the min and max values. The number of sampled glaciers is denoted above each box. Asterisks mark box pairs where the difference between median values is statistically significant. **b** Percentage change in velocity between 2000 m and 500 m from the terminus at each individual lake-terminating (blue) and land-terminating (red) glacier. **c** Pie charts show the proportion of accelerating (dark blue, dark red) to decelerating (light blue, light red) glaciers within each ice sheet region.

**Table 1 | Statistical comparison of differences between the median velocity of lake-terminating and land-terminating glaciers at each flowline distance, using a Mann-Whitney U test for difference and a significance threshold of $p$ = 0.05 (highlighted in bold)**

| Distance up-ice ($m$) | Test for difference between lake- and land-terminating glacier velocities ($p$-value) | | | | |
|---|---|---|---|---|---|
| | **2015** | **2016** | **2017** | **2018** | **2019** |
| 500 m | **<0.001** | **<0.001** | **<0.001** | **<0.001** | **<0.001** |
| 2000 m | **<0.001** | **<0.001** | **<0.001** | **<0.001** | **<0.001** |
| 3500 m | **<0.001** | **<0.001** | **0.0012** | **<0.001** | **<0.001** |
| 5000 m | 0.0634 | **0.0417** | 0.0599 | **0.0404** | **0.0311** |
| 6500 m | 0.1295 | 0.1869 | 0.2111 | 0.1347 | 0.1701 |
| 8000 m | 0.8319 | 0.7550 | 0.9436 | 0.8093 | 0.9900 |
| 9500 m | 0.4001 | 0.5510 | 0.5289 | 0.5510 | 0.5660 |

($n$ = 54, median deceleration −29%, maximum −82%) and 44% accelerated longitudinally ($n$ = 43, median 22%, maximum 160%). Further, we find that these down-ice accelerations are greater than those exhibited by land-terminating glaciers by as much as 3.2 times. The three outlet glaciers with the most substantial longitudinal accelerations show velocity increases of 160% (ID 50), 151% (ID 41) and 133% (ID 32), respectively (Fig. S3).

The proportion of lake-terminating glaciers that accelerate toward their termini to those that decelerate towards their termini is largely consistent between the SW, NW and NE ice sheet sectors, which collectively contain 74% ($n$ = 75) of the total sample. In these regions, between 46% and 50% of lake-terminating glaciers (average 48%, $n$ = 35) display towards-terminus accelerations. In the CW and southeast (SE) sectors, three of four glaciers and two of three glaciers accelerate down-ice, respectively, but in the NO sector, only three of 11 glaciers accelerate towards their termini. Land-terminating glaciers near-consistently decelerate down-ice (96%, $n$ = 98). Land-terminating glaciers exhibiting down-ice accelerations are in the SW ($n$ = 1 of 49) and NE ($n$ = 3 of 29).

**Comparison to lake area**

In addition to analysing the longitudinal velocity profile of all lake-terminating glaciers, we assessed the relationship between glacier velocity and ice-marginal lake area ($km^2$). Across the 102 glaciers studied, 34% flow into lakes between 1 and 5 $km^2$, 23% into lakes between 5 and 10 $km^2$, 13% into lakes between 10 and 15 $km^2$, 16% into lakes between 15 and 30 $km^2$, and 14% into lakes between 30 and 88 $km^2$.

Figure 2a shows the flowline velocities, extending up to 5000 m inland, of each individual glacier ordered vertically by lake area. Overall, glacier velocity appears to increase with lake area such that the fastest glaciers enter the largest lakes. Glaciers flowing into lakes larger than 30 $km^2$ ($n$ = 14) exhibit a median velocity (calculated between 500 and 5000 m) which is 40% faster than those flowing into lakes between 1 and 5 $km^2$ ($n$ = 35). However, the relationship between glacier velocity and lake size appears to be complex, with clusters of faster-flowing glaciers occupying medium sized lakes and slower glaciers occupying lakes around the 25$^{th}$ percentile in area.

This pattern is reflected in Fig. 2b, which shows the percentage change in velocity between 2000 m and 500 m along the flowline of each glacier, again ordered by lake area. Glaciers occupying the smallest lakes typically decelerate towards their termini, whilst a high number of glaciers which terminate in the medium-sized and largest lakes accelerate towards their termini (see also Fig. S4). As a result, median change in velocity is 130% greater at glaciers with lakes larger than 30 $km^2$ than at those with lakes less than 5 $km^2$ ($p$ = 0.011), but peaks in glacier acceleration are seen throughout the range of lake sizes. We find no significant difference between down-ice velocity changes at glaciers flowing into lakes >30 $km^2$ and those with lakes in each of the three remaining lake area categories (Table S2).

**Discussion**

Our results suggest that IMLs modify outlet glacier flow across the GrIS. Longitudinal velocity profiles reveal significantly faster flow at glaciers terminating in lakes than on land, especially near the terminus ( ~ 500 m up-ice), where median velocity is ~3.3 times higher. Although this enhancement effect diminishes with distance inland, elevated velocities persist for ~3.5 km. Additionally, 44% of the lake-terminating glaciers in our sample show a distinct towards-terminus acceleration, indicative of extensional flow[20]. Together, these observations point to a dynamic coupling between IMLs and GrIS outlet glaciers which extends well beyond the calving front.

IMLs typically modify glacier behavior by altering subglacial water pressures, which can enhance basal sliding, and by facilitating calving, which reduces longitudinal resistive stresses at the terminus[12,19]. Such perturbations can extend for several kilometers up-ice to increase inland flow velocities, especially where glaciers sit on retrograde slopes[17]. These mechanisms have primarily been observed at mountain glaciers[14,16], but are likely to occur at ice sheet margins, particularly in west Greenland where many glaciers exhibit alpine-style hydrology[24]. Indeed, in the first comparison of contemporary dynamics at neighbouring Greenlandic lake- and land-terminating outlet glaciers, Holt et al. (2024)[22] observed flow velocities 1.2 to 3.4 times faster at lake-terminating Isortuarsuup Sermia than land-terminating Kangaasarsuup Sermia (2 km up-ice), as well a sustained towards-terminus acceleration. Our pan-GrIS results are broadly consistent with these findings; we see median velocities 1.7 times greater at lake-terminating glaciers than at terrestrial glaciers (2 km up-ice). However, whilst Holt et al. (2024)[22] recorded enhanced flow velocities up to 15 km inland, significant differences between the median velocity of all lake- and land-terminating glaciers sampled here are limited to the lowermost ~3.5 km of the glacier trunks. Furthermore, we find that only 44% of glaciers exhibit a down-ice acceleration like that observed at Isotuarsuup Sermia. This discrepancy reflects large heterogeneity in the longitudinal flow regimes of lake-terminating glaciers across the ice sheet, wherein glaciers with velocities modified to the extent seen at Isortuarsuup Sermia are found to be amongst the most sensitive to lake forcing.

Variability in the influence of IMLs on GrIS outlet glaciers can be expected, since the capacity for lakes to induce force imbalances at the terminus depends on a series of localised factors[25]. Elsewhere, lake effects vary at a sub-regional scale, for instance in relation to ice thickness[16], but also evolve over time as lake volume changes alongside glacier retreat through a topographic basin[26–28]. In this study, since bathymetry data is currently limited, we use lake surface area as an imperfect proxy for volume. If larger lakes are assumed to be at a more advanced stage of development[29], our results suggest that the maturity of a lake affects its potential to influence outlet glacier velocity. Glaciers terminating in the largest, most established lakes are considerably faster and more likely to accelerate towards their termini than those occupying the smallest lakes (Fig. 2). However, the most significant longitudinal velocity accelerations occur into medium-sized lakes, implying that once an IML reaches a certain depth or age, or indeed when a glacier begins to retreat out of a lake basin over a prograde slope, it perpetuates a towards-terminus velocity increase but no longer enhances this acceleration over time. This non-linear relationship likely reflects a complex set of glacier-lake interactions[21] which depend on temporally evolving, glacier-specific factors such as terminus ice thickness, lake water depth, lake thermal regime, terminus morphology, and basal effective pressure[19,30].

Whilst our results reveal a widespread and persistent glacier response to IMLs, the mechanisms driving accelerated flow remain unclear. We expect that this behaviour can be partly attributed to buoyancy forces at the glacier front, governed by the relationship between ice thickness and water depth, which reduce effective pressure and thereby lower basal drag[31]. Enhanced velocities are also likely influenced by the debuttressing effect of calving, which is controlled by local factors such as ice-cliff geometry, lake water temperature, and ice buoyancy[15,32,33]. Beyond lake effects, the observed velocity differences may also arise from contrasting bed topographies, with

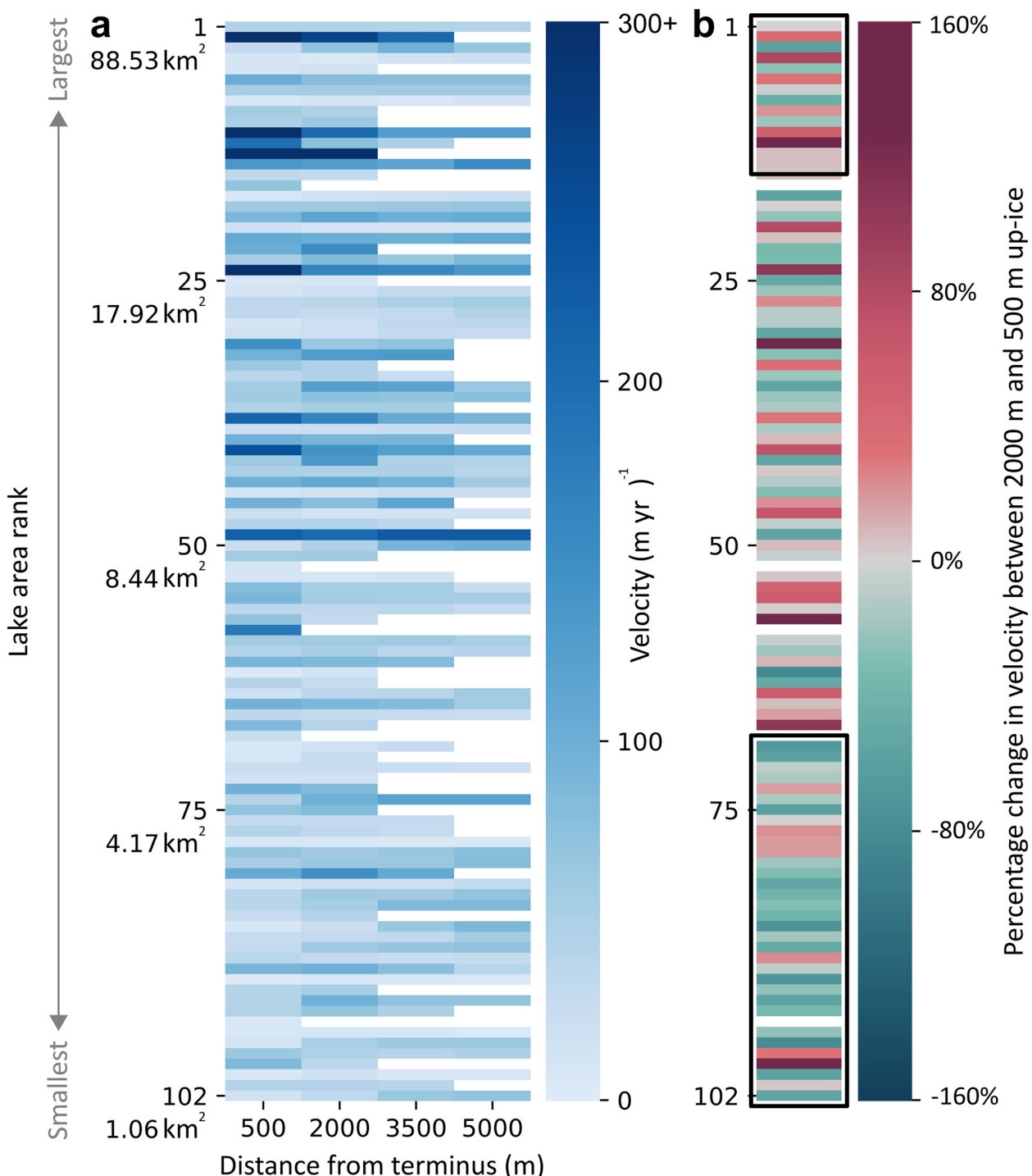

**Fig. 2 | Comparison between glacier velocity characteristics and lake area.**
**a** Longitudinal velocities and (**b**) percentage change in velocity between 2000 m and 500 m along the flowline at each individual lake-terminating glacier, ordered by lake area. One row is representative of one glacier. White pixels in both panels relate to no-data values. Black rectangles in (**b**) define glaciers with lakes ≤ 5 km² and > 30 km².

lake-terminating glaciers more likely to occupy retrograde beds and land-terminating glaciers typically lying on prograde beds. Future work that investigates these processes, for instance by disentangling the relative influence of terminus ablation and subglacial hydrology on surface velocity, will be essential for representing lake-terminating outlet glaciers in numerical ice sheet models. We therefore highlight the need for detailed analyses of glacier dynamics alongside concomitant lake conditions to better understand how GrIS outlet glaciers may respond to future lake development. Likewise, studies which quantify lake effects over interannual to

decadal timescales would provide insight into how the velocity enhancements described here evolve over time.

At present, the volume of ice draining into lakes represents only a small percentage of total ice discharge[13]. However, since IMLs are predicted to grow in number and size, and some marine-terminating glaciers may retreat onto land, it is anticipated that lake-terminating glaciers will increasingly contribute to GrIS mass loss over the coming decades[10,13]. This is especially true if a large number of glaciers are thinning dynamically in response to lake effects. Our findings carry implications for mass balance estimates

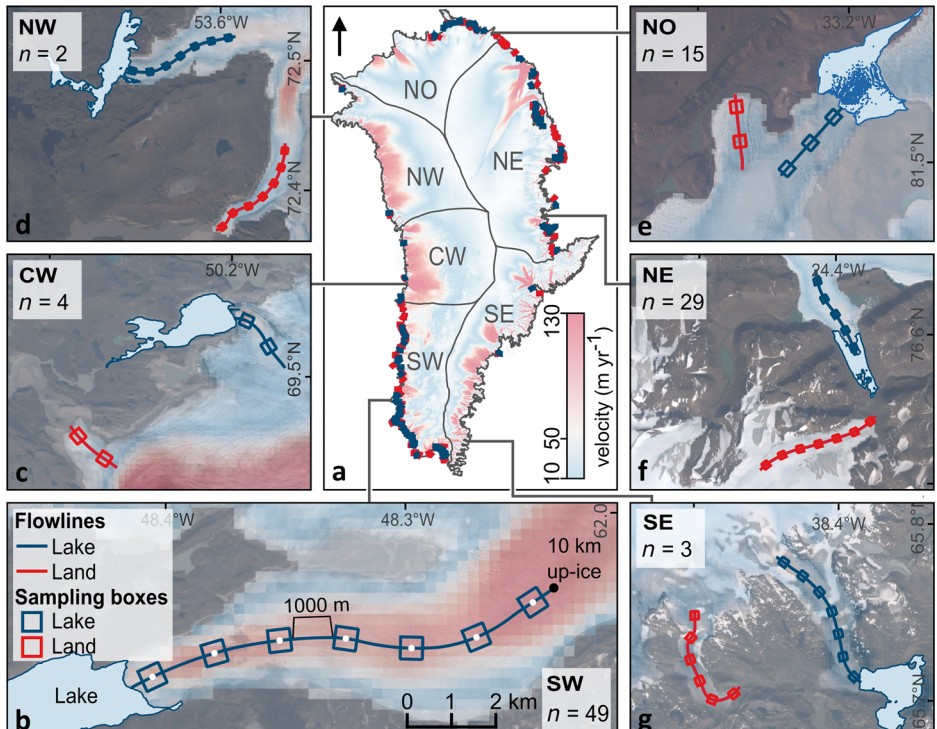

**Fig. 3 | Glacier locations and examples. a** Map of glacier locations and ice sheet regions overlaid on the ITS_LIVE 2017 annual velocity mosaic. **b** An example lake terminating outlet glacier in southwest Greenland with the digitised flowline and sampling boxes. Background is Sentinel-2 imagery from August 2019. **c–g** show example lake- and land-terminating study glaciers from the (**c**) central-west (CW), (**d**) northwest (NW), (**e**) north (NO), (**f**) northeast (NE), and (**g**) southeast (SE) regions. Numbers denote the number of lake- and land-terminating glaciers sampled within each region.

derived from the input-output method, which typically account for ice flux only at the marine margins of the GrIS[34], but perhaps more importantly, they also suggest that present and future mass loss may be underestimated if glacier-lake interactions are omitted from model scenarios.

## Methods
### Glacier selection
To assess the impact of ice-marginal lakes on outlet glacier velocity, we selected glaciers across all six regions of the GrIS defined by Mouginot et al. (2019)[4]. IML locations and extents were obtained from the How et al. (2021)[8] Greenland-wide inventory for the year 2017, which is available as a set of vector polygons derived from 10 m resolution Sentinel-1 SAR imagery and ArcticDEM Version 3[35]. Herein, we focus our analysis on ice sheet outlet glaciers, identified as distinct flow fields at the ice sheet margin, that terminate in lakes larger than 1 km². This threshold favours lakes of a significant size and ensures that any given lake – ice contact margin is greater than the spatial resolution of the velocity dataset (120 m). In turn, the final selection comprises a set of 102 lake-terminating outlet glaciers, which collectively host 46% of the total mapped area of lakes abutting the ice sheet margin. Note that smaller sample numbers in CW, NW and SE reflect a lower average lake area ( < 1 km²) in these regions. To provide a basis for comparison, a corresponding set of land-terminating glaciers was chosen within each region by selecting the nearest land-terminating glacier with similar dimensions to each lake-terminating glacier (Sup. Note 1; Fig. 3). To ensure that glacier characteristics other than terminus environment were consistent across the two sample populations, the average ice surface elevation, slope and aspect within each flowline box on each glacier was extracted from ArcticDEM (10 m pixel resolution[36]). When compared between lake- and land-terminating glaciers, we found no significant ($p < 0.05$) differences in elevation, aspect or slope, except at 500 m up-ice, where glaciers flowing onto land are significantly steeper at the surface than those that flow into lakes ($p < 0.01$) (Fig. S5; Table S3).

### Ice surface velocities
Ice surface velocities were obtained from the NASA Making Earth System Data Records for Use in Research Environments (MEaSUREs) Inter-mission Time Series of Land Ice Velocity (ITS_LIVE) velocity magnitude Version 2 dataset[37,38]. ITS_LIVE V2 data are derived using the auto-RIFT feature tracking software applied to measure pixel displacement between Landsat-4, -5, -7 and -8 optical image pairs, processed at a spatial resolution of 120 m. Each annual velocity mosaic is the error-weighted average of all image-pair velocity scenes with centre dates which fall within the same calendar year[39]. To match the timestamp of the IML inventory, we focus our analysis on ice surface velocity during 2017. Additional velocity data from 2000, 2015, 2016, 2018, and 2019 were assessed to verify that the results produced for 2017 are broadly consistent with, and therefore representative of, trends observed in other years (Fig. S6).

Following similar studies extracting spatially or temporally consistent datasets of longitudinal patterns in glacier velocity (e.g.[40,41]), velocity measurements were sampled at multiple evenly spaced locations along each glacier trunk. To determine these locations, a flowline for every glacier was manually and systematically digitised to follow the line of fastest flow as observed in the ITS_LIVE 2017 velocity data, guided by surface features where visible in the MEaSUREs Greenland satellite image mosaics[42]. Each flowline extends inland from a glacier terminus, either until the head of a glacier trunk is reached at the point of flow initiation or flow divergence, or for a maximum distance of 10 km (Fig. 3). To enable comparison between individual glaciers, sampling boxes were centred around points spaced at 1500 m intervals along the flowlines between 500 m and 9500 m from the glacier termini. This produced a chain of up to seven sampling boxes for each glacier, depending on the flowline length. Box dimensions of 500 m by 500 m were chosen to prevent overlap with the lateral margins of the narrowest glacier whilst allowing coverage of an average of 17 120 × 120 m pixels, thereby avoiding point sampling from individual pixels with relatively high levels of associated error. Results produced using this systematic sampling approach are robust to changes in box dimensions, when tested across

glaciers of various regions and flow regime (Fig. S7). Annual velocity values were then extracted as the mean pixel value within each sampling box. For analysis, aggregate velocity profiles representing all sampled lake- and land-terminating glaciers were created. For data relating to the year 2000, this is limited to velocity sampled at 500 m up-ice. Error estimates for each box-sampled velocity measurement were calculated conservatively as the maximum per-pixel error value within each box, derived from the ITS_LIVE velocity magnitude error mosaics. The median of these maximum error values was $0.5 \text{ m yr}^{-1}$, with 91% of boxes displaying maximum per-pixel error values below $2 \text{ m yr}^{-1}$. Since the results are reported as aggregate velocities values calculated across multiple locations, and observed velocities are typically several orders of magnitude greater than the calculated error, we report velocity values to the nearest metre per year.

We use the Mann-Whitney U test to statistically assess the similarity of lake- and land-terminating glacier velocities and topographic characteristics, and lake-terminating glacier velocities across lake area bins. This test was chosen for its suitability for continuous data within two independent samples with skewed distributions.

### Reporting summary

Further information on research design is available in the Nature Portfolio Reporting Summary linked to this article.

### Data availability

Ice-marginal lake outlines are available from How et al. (2021) https://catalogue.ceda.ac.uk/uuid/7ea7540135f441369716ef867d217519 and annual gridded velocity data from the NASA MEaSUREs ITS_LIVE V2 Regional Glacier and Ice Sheet Velocity dataset (Gardner et al., 2019). Shapefiles of glacier flowlines and sampling boxes and all data derived from these, are available via the UK Polar Data Centre: https://doi.org/10.5285/fa7887ea-5c4e-4d7a-b39f-456792f34b37.

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

## Acknowledgements
C.M.H. is supported by a UKRI Natural Environment Research Council PhD studentship (NE/S007458/1). We thank the editors Prof. Shin Sugiyama and Dr. Alireza Bahadori for handling the paper, as well as Dr. Nicole Abib and two anonymous reviewers for their helpful and constructive feedback.

## Author contributions
C.M.H designed the study with the help of M.W.S, D.J.Q, J.L.C, and L.T. C.M.H acquired and analysed the data, drafted the manuscript and produced the figures. All authors contributed to the interpretation of the data and revisions to the manuscript text.

## Competing interests
The authors declare no competing interests.
