## [Transparent Peer Review file · Communications Earth & Environment]

Ice-marginal proglacial lakes enhance outlet glacier velocities across Greenland

Corresponding Author: Ms Connie Harpur

This manuscript has been previously reviewed at another journal. This document only contains information relating to versions considered at Communications Earth & Environment.

Version 0:

Decision Letter:

Dear Ms Harpur,

Your manuscript titled "Ice-marginal Lakes Enhance Outlet Glacier Velocities Across Greenland" has now been seen by 3 reviewers, whose comments are appended below. You will see that they find your work of some potential interest. However, they have raised substantial concerns that must be addressed. In light of these comments, extensive revisions will be required before we can further consider the manuscript for publication. We would, however, be interested in considering a revised version that fully addresses these serious concerns. Specifically, a revised manuscript must:

1. Compellingly demonstrate the robustness of your conclusions by incorporating a temporal analysis to determine whether the observed velocity differences represent a recent change or a persistent feature of the ice sheet.
2. Fully justify the use of the lower-resolution and older dataset by evaluating their impact on the precision of velocity measurements for narrow land- and lake-terminating glaciers.
3. Conduct a sensitivity analysis and provide a synthesis of the underlying physical mechanisms and local variables to demonstrate the robustness of the parameters and clarify the drivers of glacier acceleration.

We hope you will find the reviewers' comments useful as you decide how to proceed. If additional work allows you to either incorporate or refute these criticisms, we will be happy to look at a substantially revised manuscript. If you choose to take up this option, please either highlight all changes in the manuscript text file, or provide a list of the changes to the manuscript with your responses to the reviewers.

When resubmitting, please provide a point-by-point response to the reviewers' comments. Please submit your responses as a separate file, distinct from your cover letter where you can add responses to the Editors' comments that you do not want to be made available to the reviewers. Word files are preferred. We recommend that any figures, tables or graphs that are included in the response to reviewers are also included in the main article or Supplementary Information.

If the revision process takes significantly longer than three months, we will be happy to reconsider your paper at a later date, as long as nothing similar has been accepted for publication at Communications Earth & Environment or published elsewhere in the meantime.

Please use the following link to submit your revised manuscript, point-by-point response to the reviewers' comments with a list of your changes to the manuscript text (which should be in a separate document to any cover letter), a tracked-changes version of the manuscript (as a PDF file) and any completed checklist:

Link Redacted

Please do not hesitate to contact us if you have any questions or would like to discuss the required revisions further. Thank you for the opportunity to review your work.

Best regards,

Shin Sugiyama, PhD
Editorial Board Member
Communications Earth & Environment
orcid.org/0000-0001-5323-9558

Alireza Bahadori, PhD
Senior Editor
Communications Earth & Environment
Consulting Editor
Communications Sustainability

EDITORIAL POLICIES AND FORMAT

If you decide to resubmit your paper, please ensure that your manuscript complies with our editorial policies and complete and upload the checklist below as a Related Manuscript file type with the revised article:

- Behavioural and social science
- Ecological, evolutionary & environmental sciences
- Life sciences

For your information, you can find some guidance regarding format requirements summarized on the following checklist: (<https://www.nature.com/documents/commsj-phys-style-formatting-checklist-article.pdf>) and formatting guide (<https://www.nature.com/documents/commsj-phys-style-formatting-guide-accept.pdf>).

REVIEWER COMMENTS:

Reviewer #1 (Remarks to the Author):

Please find the comments in the attachment.

Reviewer #2 (Remarks to the Author):

Summary:

In this paper, the authors utilize a combination of two pre-existing datasets to contrast the velocity differences between glaciers with ice marginal lakes and those that are land-terminating. The study is built on an inventory of Greenland-Wide Ice Marginal Lakes produced from Sentinel 1 imagery in 2017 and then contrasted with a 2017 annual mosaic of ice surface velocities from the NASA MEaSUREs ITS_LIVE dataset. Statistical analyses between the two datasets reveal that glaciers with ice marginal lakes are almost 3 times as fast as their nearby land-terminating counterparts, and therefore greater attention should be paid to these types of glaciers in models of ice sheet evolution.

The paper is very well-written, the figures are easy to understand, and the manuscript provides novel insight into lake-terminating glacier dynamics. I just have a few main comments which I include below.

Main Points:

1. My main piece of feedback is on the use of the terminology "ice marginal lake" vs. "proglacial lake" throughout your

manuscript. The methods section specifies that you are only looking at glaciers that terminate in lakes, but the term “ice marginal” in your manuscript title and elsewhere in the text implies that you are also looking at glaciers that have ice-dammed lakes adjacent to them but may have no lake at the terminus. Since the focus of your paper is really on proglacial lake-terminus interactions, I suggest rephrasing throughout.

2. I appreciate the authors use of pre-existing datasets to provide new insight into ice sheet evolution. While other manuscripts have contrasted glacier-average differences in velocities between land-terminating and lake-terminating glaciers, this manuscript dives deeper to examine whether these glaciers accelerate or decelerate near their termini. A main point of your paper that seems overshadowed to me is that only 32% of lake terminating glaciers accelerate longitudinally towards the terminus, whereas prior studies cited (i.e., Holt et al. 2024) have emphasized the along-flow acceleration of glaciers with proglacial lakes. Your analysis of proglacial lakes of various sizes shows that there is more nuance to this than previously thought, and this point could be emphasized more throughout the paper. The cryosphere community’s bias on studying the biggest, fastest retreating glaciers can often overlook nuanced findings like this that show glaciers can behave quite heterogeneously depending on their size, all of which are contributing to mass loss of the Greenland Ice Sheet.

3. Have you considered the impact of ice velocity seasonality on your results? As you mention in your discussion, the ice velocity of lake-terminating glaciers is influenced by several processes, such as subaqueous melt and subglacial hydrology, that all vary seasonally. How might using an annual velocity mosaic compared to the closest velocity data in time to the lake outline impact your findings?

4. While this study focuses on the pan-Greenland patterns in lake-terminating glacier ice velocity, I think it could be useful to include your results from individual glaciers in the supplementary material, while also pointing out specific examples in your results and discussion sections (see Slater et al. 2022 as an example: <https://www.nature.com/articles/s41561-022-01035-9>). This manuscript sets the stage for a harder look into glacier-lake interactions, so providing your dataset and specific examples will facilitate the future study of the dynamics that are driving the broadscale patterns you observe.

5. I would recommend adding a section to the end of your methods on the statistical tests you performed. Which tests did you choose to examine your data, why, and what assumptions go into them?

Line by line comments:

L32: “More than half of this mass loss occurs via outlet glaciers” – where else would this mass loss occur? Do you mean marine-terminating outlet glaciers in comparison to land-terminating? Or dynamic loss via calving vs. surface melting? Specify.

L51: Should “ablation” be “frontal ablation”?

L82: Can you be more specific about the “implications for the inclusion of IMLs in ice sheet models”. For those unfamiliar with how they are currently implemented or not, what specifically about your findings will be relevant for ice sheet models?

L92: Should “negligible differences” be “insignificant differences”? Or how do you define negligible

L99: Should “at lake- and land-terminating” be “between lake- and land-terminating”?

L103: Specify what IQR means upon its first use

L106: Why are there such limited sample numbers outside of the SW, N, and NE regions? Does it have to do with the datasets and size criteria you used, or just the number of glaciers with ice marginal lakes in those regions? Could be helpful to specify.

L120: Along the lines of my comments above, it could be useful to name these three glaciers that show the largest acceleration for subsequent studies.

L129: It would be helpful to put a description of the error bars in Figure 1a into the Figure 1 caption

L153: Should “occupy the largest lakes” be “enter the largest lakes”?

L157: Add a label for the 25th percentile in area to Fig. 2 for easier interpretation

L165: How did you choose the lake size categories for looking at the lake-area vs. down-ice velocity changes (Table S2)? If these size categories are shifted, how does this change your findings of significant vs. insignificant? I’d be curious to see a line plot of percentage change in velocity vs. lake size to discern how ice acceleration depends on lake size.

L181: Add citation for the 5km inland distance surpassing the grounding line location

L194: Specify what Isotuarsuup Sermia is flowing 1.2-3.4x faster than

L196: Change “lacustrine glaciers” to “lake-terminating glaciers” to match previous wording

L208-214: Are there any lakes in your dataset that you could look into for more of a case study on these claims? Somewhere you could create a time series of lake-size, overall glacier velocity longitudinal profile, and near terminus acceleration? I suspect that the relationship between velocity acceleration and lake size has more to do with a ratio of terminus thickness to lake depth, rather than lake age. For example, if a glacier terminates in a very large lake, but is just about to retreat onto land, I imagine it would have a much different velocity profile compared to a glacier terminating in the same size lake that still has substantial subaqueous contact. Either diving into a case study, or adding this caveat would help strengthen your claims.

L249: Specify what dimensions you used to find a comparable land-terminating glacier. Length, width, height, ice flux? Also note that in Supp Note 1 you refer to section 2.1, which I think should be 4.1

L287: Recommend re-ordering the labeling of subfigures here so they go in clockwise order and for example, panels b and c are next to each other.

-Nicole Abib

Reviewer #3 (Remarks to the Author):

This manuscript presents a comprehensive, pan-Greenland assessment of how ice-marginal lakes (IMLs) influence outlet glacier dynamics. The authors compared longitudinal velocity profiles of 102 lake-terminating glaciers with a matched set of land-terminating glaciers across all six regions of the Greenland Ice Sheet, using 2017 velocity data from NASA's ITS_LIVE dataset and lake inventories from How et al. (2021).

This is a well-written paper with a clear and logical structure. The research question is timely and relevant, the methodology is generally sound, and the results provide important new insights into glacier-lake interactions at the ice sheet scale. The manuscript would make a valuable contribution to Communications Earth & Environment. However, I have several suggestions for improvement before publication.

(1) In the method section 4.1, I would suggest to give a statistical characterization of lake-terminating and land-terminating glacier samples. Currently, the manuscript states that land-terminating glaciers were selected based on "similar dimensions," but without quantitative metrics, it's difficult to evaluate how well-matched these populations are. I recommend adding a table or figure in Section 4.1 that presents the distribution of key glaciological parameters for both lake-terminating and land-terminating glaciers, including: Glacier area, Elevation range (min, max, median), slope, Aspect/orientation, etc. This would allow readers to assess whether the two populations are truly comparable and whether any differences in velocity could be partially explained by systematic differences in glacier geometry or topographic setting.

(2) In the method section 4.2, the manuscript states that flowlines were "systematically digitised to follow the line of fastest flow as observed in the ITS_LIVE 2017 velocity data, guided by surface features where visible." However, it is unclear whether this digitization was performed manually or through an automated algorithm. This distinction is critical for assessing reproducibility and potential bias in the analysis.

(3) In the method section 4.2, the choice of 500 m × 500 m sampling boxes spaced at 1500 m intervals appears reasonable but lacks justification. The scientific robustness of these parameters needs to be demonstrated, as different choices could potentially yield different results. I would recommend conducting a sensitivity analysis using a subset of representative glaciers (e.g., 5-10 glaciers spanning different sizes, velocities, and regions). Present the results as a supplementary figure or table demonstrating that the main conclusions are robust to these methodological choices. This would strengthen confidence in the findings and address potential reviewer concerns about parameter selection.

(4) The reference list does not conform to the formatting requirements of Communications Earth & Environment. The journal follows a numbered citation system with a specific format for references. Please carefully review the journal's author guidelines and reformat all references accordingly.

(5) Some abbreviations should be given in full name when they appear first time in the main text, e.g. IQR in Line 103; SW, CW, NW and NE in Line 122;

(6) Add (c) in Line 137; N or NO should be consistent in Line 292; four 240 m² pixels, do you mean four 240m * 240m pixels?

** Visit Nature Portfolio's author and referees' website at www.nature.com/authors for information about policies, services and author benefits**

Communications Earth & Environment is committed to improving transparency in authorship. As part of our efforts in this direction, we are now requesting that all authors identified as 'corresponding author' create and link their Open Researcher

and Contributor Identifier (ORCID) with their account on the Manuscript Tracking System prior to acceptance. ORCID helps the scientific community achieve unambiguous attribution of all scholarly contributions. You can create and link your ORCID from the home page of the Manuscript Tracking System by clicking on 'Modify my Springer Nature account' and following the instructions in the link below. Please also inform all co-authors that they can add their ORCIDs to their accounts and that they must do so prior to acceptance.

Version 1:

Decision Letter:

Dear Ms Harpur,

Your revised manuscript titled "Ice-marginal proglacial lakes enhance outlet glacier velocities across Greenland" has now been seen by our reviewers, whose comments appear below. In light of their advice we are delighted to say that we are happy, in principle, to publish a suitably revised version in Communications Earth & Environment.

We therefore invite you to revise your paper one last time to address the remaining concerns of our Reviewer #1. At the same time we ask that you edit your manuscript to comply with our format requirements and to maximise the accessibility and therefore the impact of your work.

EDITORIAL REQUESTS:

****Please take care to match our formatting and policy requirements. We will check revised manuscript and return manuscripts that do not comply. Such requests will lead to delays. ****

SUBMISSION INFORMATION:

OPEN ACCESS:

Communications Earth & Environment is a fully open access journal. Articles are made freely accessible on publication. For further information about article processing charges, open access funding, and advice and support from Nature Portfolio, please visit <https://www.nature.com/commsenv/open-access>

Link Redacted

Best regards,

Shin Sugiyama, PhD
Editorial Board Member
Communications Earth & Environment
orcid.org/0000-0001-5323-9558

Alireza Bahadori, PhD
Senior Editor
Communications Earth & Environment
Consulting Editor
Communications Sustainability

REVIEWERS' COMMENTS:

Reviewer #1 (Remarks to the Author):

The authors have addressed all comments raised in the first round of review. I am satisfied with the revised manuscript and have no further major comments.

I have only a few minor suggestions for the authors to consider to further polish the discussion and clarify limitations:

1. Lake area vs. lake depth

The analysis in the manuscript establishes a relationship between glacier velocity and lake area (Figure 2). However, the physical mechanisms driving flow acceleration (buoyancy/flotation and thermal undercutting) are primarily functions of water depth rather than surface area. While I understand that bathymetry data is not available, it would be beneficial to explicitly acknowledge in the discussion that "lake area" is an imperfect (a compromise) proxy for depth. This would help explain the "complex" non-linear relationship observed, where medium-sized lakes also induce significant acceleration.

2. Bed topography controls

The revision clarifies that surface slope was used to match the control group glaciers. However, it is likely that many land-terminating glaciers reside on prograde beds, while lake-terminating glaciers lie on retrograde (overdeepened) beds. Since bed topography was not explicitly controlled for, acknowledging this as a potential contributing factor to the observed velocity differences (beyond just the "lake effect") would add rigor to the interpretation.

Reviewer #2 (Remarks to the Author):

The authors have addressed thoroughly the points I raised in my initial review. The paper remains very well-written and presents a thorough and valuable dataset.

-Nicole Abib

Reviewer #3 (Remarks to the Author):

The author(s) have done a good job to revise the manuscript according to the reviewers' comments and suggestions. I have no further comments. In my view, the manuscript can be accepted now.

** Visit Nature Portfolio's author and referees' website at <http://www.nature.com/authors> for information about policies, services and author benefits**

Ice-marginal proglacial lakes enhance outlet glacier velocities across Greenland

Response to reviewers

Summary

We sincerely thank the three reviewers for their helpful and constructive comments and suggestions. Please find our point-by-point responses printed in blue below, with new text from the updated manuscript quoted in *italics*. This response accompanies a revised version of the manuscript (Anon_Manuscript_Comms_Earth_Env_Revised) and supplement (Anon_Supplementary_Info_Comms_Earth_Env_Revised), and revised versions with tracked changes. Line numbers below refer to the tracked changes version.

Reviewer #1 (Remarks to the Author):

This manuscript presents a pan-Greenland analysis of the influence of ice-marginal lakes (IMLs) on the dynamics of outlet glaciers. The authors compare the longitudinal velocity profiles of 102 lake-terminating glaciers with a corresponding set of 102 land-terminating glaciers using publicly available satellite data from 2017. The primary findings are that lake-terminating glaciers are significantly faster than their land-terminating counterparts (by ~295% at the terminus), with this effect extending approximately 5 km inland. Furthermore, about one third of lake-terminating glaciers exhibit acceleration towards their terminus, a behaviour that is rare in land-terminating glaciers. The study concludes that IMLs exert a crucial control on Greenland Ice Sheet outlet glacier dynamics, a factor that should be incorporated into future ice sheet models.

General Comments

This study addresses a significant knowledge gap in our understanding of the role of IMLs in the Greenland Ice Sheet. The expansion of IMLs is a well-studied result of climate change, but their integrated, ice-sheet-wide impact on glacier dynamics has remained unquantified. The authors utilized a public lake inventory and ice velocity dataset, providing a large-scale quantification of this effect. The study is novel in its scope, and the findings have clear implications for the ice sheet modelling community.

However, the study in its current form has several limitations related to its temporal scope and methodological choices that raise significant questions about the robustness and context of the conclusions. While the key finding is likely correct, the issues detailed in the following sections must be addressed through a major revision before the manuscript can be considered for publication.

Major comments:

1. Lack of historical context and temporal analysis

The study's most significant limitation is its reliance on a spatial "snapshot" analysis. While the authors use 2015–2019 data to verify their 2017 results, this recent focus leaves a critical question unanswered: is the observed velocity difference a recent phenomenon, or has it been a long-standing feature of the ice sheet? The introduction

states that IMLs “have the propensity to alter glacier dynamics and enhance rates of ice mass loss,” but it is unclear if the lake-terminating glaciers studied were already flowing faster than their land-terminating counterparts before any recent enhancement. Therefore, a temporal analysis is necessary to support the claim of “enhancement,” which currently implies a temporal speed-up that the study does not show.

While I do not expect a full temporal analysis within the scope of a revision, it is reasonable to ask the authors to perform a preliminary analysis for an earlier period (e.g., a year or block of years before 2010) using the available ITS_LIVE velocity data. This would provide a first-order test of whether their conclusion holds true for a different climatic period. Additionally, the authors should clearly state that their study demonstrates an “enhanced velocity state” and a different spatial velocity profile in the presence of lakes but does not track glacier velocity changes through time. This discussion should frame the study as a crucial baseline that powerfully motivates future research into the temporal co-evolution of IMLs and glacier dynamics.

Thank you for this suggestion. We agree that presenting velocity data for an earlier year in a different climatic period provides important temporal context and have therefore performed analysis of lake-terminating and land-terminating glacier velocities during the year 2000.

To do this, we firstly verified lake presence using the MEaSURES GIMP 2000 Image Mosaic, which is constructed from MODIS imagery from the years 1999 to 2002. Of the 102 lake-terminating glaciers sampled using the 2015-2019 velocity data, 16 were found to be land-terminating in 2000 and are therefore excluded from this new analysis. Neighbouring land-terminating glaciers from the original sample ($n = 102$) were also excluded from analysis to produce even sample sizes of $n = 86$ lake-terminating and land-terminating glaciers.

Since most of the sampled glaciers have retreated since 2000, our original flowlines (and therefore sampling boxes) are unsuitable for application to velocity mosaics from earlier years. Drawing new flowlines for the year 2000 would be a significant undertaking, so, with a view to investigating whether the relationship between lake- and land-terminating glaciers in 2000 is consistent with the post-2010 data, we chose to focus on comparing lake- and land-terminating glacier velocities in the year 2000 at 500 m distance from glacier termini.

To extract these year 2000 velocities, we measured 500 m up-ice from the terminus as seen in the MEaSURES GIMP mosaic, and sampled the year 2000 ITS_LIVE Version 2 velocity and error mosaics as the average pixel value within a 500 m circumference circle. This allowed us to directly compare the velocity of lake-terminating and land-terminating glaciers at 500 m from the terminus during the year 2000. These results are presented in Fig. S2; the most salient point is that, in the year 2000, the median lake-terminating glacier velocity was 76% faster than the median land-terminating glacier velocity (at the 500 m distance up-ice location).

Extracting full velocity profiles for each glacier was prohibited by the lack of flowlines for 2000. Nevertheless, the analysis undertaken demonstrates that the key relationship between terminus environments in the 2015-2019 data (i.e. generally faster velocities at lake-terminating glaciers) was also present pre-2010. It appears that this relationship has evolved over the years since 2000 (whereby the percentage difference between lake

and land glacier velocities at 500 m up-ice has changed from 76% in 2000 to 225% in 2017), suggesting a temporal evolution of glacier response to lake effects.

Given all the above, we have now added the following text to the results, and a new figure to the supplement (Fig. S2):

Lines 116 – 119 (Results): *To test whether our results are specific to the period 2015 - 2019, or instead persistent, we also analysed velocity during the year 2000 at 500 m up-ice from the termini. Like in 2015 - 2019, lake-terminating glaciers were significantly ($p < 0.01$) faster at 500 m up-ice (51 m yr^{-1}) than land-terminating glaciers (29 m yr^{-1}) during the year 2000 (Fig. S2).*

More generally, we have clarified in the introduction that the “enhancement” we refer to relates to the difference between lake- and land-terminating glacier velocities during 2017, rather than an enhancement over time, and highlighted the importance of temporal data in the discussion:

Lines 80 – 82 (Introduction): *Our results reveal a 231% increase in the terminus velocity of lake-terminating glaciers compared to land-terminating glaciers during 2017.*

Lines 259 – 261 (Discussion): *Likewise, studies which quantify lake effects over interannual to decadal timescales will provide insight into how the velocity enhancement described here evolves over time.*

2. Use of an old version of the velocity dataset

The authors have used version 1 of the ITS_LIVE velocity dataset, which has a spatial resolution of 240 m. A newer, improved version 2 has been available for over a year and has a higher resolution of 120 m. Scientific best practice dictates using the most current and highest quality data available. The research targets are land- and lake-terminating glaciers, which are relatively narrow compared to marine-terminating glaciers. Thus, the use of lower-resolution data may impact the study's precision, as a 240 m pixel size may be insufficient to accurately capture flow dynamics at the margins and terminus. The authors should justify using the older dataset. Ideally, they should be required to re-run their entire analysis using the 120 m ITS_LIVE version 2 data. This would significantly increase confidence in the precision and robustness of their quantitative results.

Thank you for highlighting this updated version of the dataset. ITS_LIVE Version 1 was the current release at the time of the original data analysis, but we have now repeated our analysis in full to update the results and manuscript using ITS_LIVE Version 2. These new results are largely consistent with the original data and therefore support our original discussion and conclusions, but we found some differences worthy of note. Specifically:

- In the original manuscript, we reported statistically significant differences between lake- and land-terminating glacier velocities at each flowline location up to 500 m inland in every sample year (2015-2017). Using ITS_LIVE Version 2 data, we find that this pattern is present for years 2016, 2018 and 2019, but that significant differences between lake- and land-terminating glacier velocities are limited to the 500 m, 2000 m and 3500 m sample boxes for years 2015 and 2017. We have amended the wording in lines 98 – 102 to reflect this change:

Lines 98 – 102 (Results): *Whilst the two groups display negligible differences in the up-ice sample boxes, the aggregated velocity profiles diverge towards the ice sheet margin such that, from 3.5 km inland, lake-terminating glaciers are both significantly ($p < 0.01$) and progressively faster than their land-terminating counterparts.*

- The original manuscript reported that 31 of 97 lake-terminating glaciers accelerated longitudinally towards their termini. Using ITS_LIVE Version 2, this number increases to 43 of 97 glaciers. We attribute this change to the finer resolution of the newest ITS_LIVE mosaics, which, as highlighted by the reviewer, can capture more detailed variability in ice sheet-marginal areas. Despite this increase in the number of accelerating lake glaciers, we still find a significant difference between the percentage change in velocity between 2000 m and 500 m from the terminus at glaciers flowing into the smallest (1 to 5 km²) and largest (15.01 to 30 km²) lakes.

This new data has resulted in amendments to values and statistics throughout the text; these changes are recorded in the *_TrackedChanges* version of the revised manuscript.

3. Insufficient investigation of physical mechanisms

The manuscript establishes a pan-Greenland correlation between IMLs and higher glacier velocities but provides little insight into the underlying physical mechanisms. While the authors identify several potential drivers in the Introduction and Discussion (including calving, reduced buttressing, and the alteration of subglacial water pressures), the analysis does not test or discuss in any depth the relative importance of these processes, leaving the "how" and "why" behind their main finding largely unanswered.

I suggest the authors expand their Discussion to provide a more thorough synthesis of the underlying physics and the key local variables that control each process (e.g., the role of lake depth in calving; the influence of bed properties on subglacial hydrology).

We are keen to avoid too much speculative analysis, since our manuscript doesn't aim to provide robust insight into these mechanisms. Indeed, data needed to investigate these processes (i.e. lake temperatures, lake depths, bed pressure, and calving records) are currently extremely sparse, and therefore we suggest that such analysis would be a helpful focus for future work. To provide some additional context to our results and expand the discussion around possible mechanisms driving the observed dynamics, we have added the following text:

Lines 246 – 255 (Discussion): *Whilst our results reveal a widespread and persistent glacier response to IMLs, the mechanisms driving accelerated ice flow remain unclear. We expect that this behaviour can be partly attributed to buoyancy forces at the glacier front, governed by the relationship between ice thickness and water depth, which reduce effective pressure and thereby lower basal drag (Benn et al., 2007). Enhanced velocities are also likely influenced by the debuttressing effect of calving, which is controlled by local factors such as ice-cliff geometry, lake-water temperature, and ice buoyancy (Liu et al., 2020; Röhl et al., 2017; Mallalieu et al., 2021). Future work that investigates these processes, for instance by disentangling the relative influence of terminus ablation and subglacial hydrology on surface velocity, will be essential for representing lake-terminating outlet glaciers in numerical ice sheet models.*

4. Ambiguity in the selection of the control group

The methodology for selecting the land-terminating control glaciers is described qualitatively as choosing the "nearest land-terminating glacier with similar dimensions." This is a critical step in the methodology, and the lack of quantitative criteria could introduce potential bias (e.g., are lake-terminating glaciers systematically located in regions with steeper slopes or different subglacial conditions that would facilitate faster flow?).

The authors should strengthen the Methods section by providing a more detailed, and if possible, quantitative description of the criteria used for pairing glaciers (e.g., ranges of width, catchment area, slope, etc.). This would increase confidence that the primary differentiating variable between the two groups is indeed the terminus environment. I also suggest the authors share the related shapefiles with the reviewers, even at the peer-review stage, to allow for verification.

Thank you for this suggestion. The sampled glaciers have not yet been identified (either at all, or specifically as lake-terminating) in existing datasets (e.g. RGI or Mankoff et al., 2020), so, given the methodological issues and high levels of error associated with computing catchment areas for Greenlandic lake-terminating outlet glaciers (as in Carrivick et al., 2022), we focus our quantitative comparison of lake- versus land-terminating glaciers on topographic variables, specifically elevation, slope and aspect.

We extracted the mean elevation, slope and aspect within each flowline box (500 m – 9500 m) on each lake and land glacier from the ArcticDEM raster of ice surface elevation (10 m resolution). For all three topographic variables, and for all distances up-ice, we find no statistical difference between the lake and land glacier populations. The reviewer might be interested to see that there is one exception, whereby land glaciers have significantly steeper surface slopes at 500 m up-ice. It is unclear why this pattern exists but, given that steeper ice surface slopes are usually associated with increased ice velocities, we note that lake glaciers are significantly faster in the terminus region than land glaciers *despite* the association between land glaciers and steeper surface slopes.

This analysis has now been added to the methods section and supplement (Fig. S5):

Lines 289 – 295 (Methods): *To ensure that glacier characteristics other than terminus environment were consistent across the two sample populations, the average ice surface elevation, slope and aspect within each flowline box was extracted from ArcticDEM (10 m pixel resolution; Porter et al., 2022) and compared between lake- and land-terminating glaciers. We found no significant ($p < 0.05$) differences in the elevation, aspect or slope of lake- and land-terminating glaciers, except at 500 m up-ice, where glaciers flowing onto land are significantly steeper at the surface than those that flow into lakes ($p = 0.0003$) (Fig. S5).*

Minor Points / Line-by-Line Comments

Title: As noted, "Enhance" could be misinterpreted as a temporal speed-up. The authors should ensure the text clarifies that the study demonstrates an enhanced velocity state.

We have clarified this within the introduction.

Citation Style: There appear to be two citation styles in the manuscript. Please check the journal requirements and use a consistent style.

The reference list and citations have been updated to conform exactly to the journal guidelines.

Line 14 (Abstract): The value "295% faster" should be clarified. The reported median values (36 m yr^{-1} and 9 m yr^{-1}) would suggest a 300% difference.

Amended according to updated results:

Lines 107 – 110 (Results): *As a result, the difference in median velocity at lake- and land-terminating glaciers is greatest within the terminus region, reaching 30 m yr^{-1} at 500 m inland where lake- and land-terminating glaciers display median velocities of 43 m yr^{-1} and 13 m yr^{-1} , respectively.*

Lines 80 – 82 (Introduction): *Our results reveal a 231% increase in the terminus velocity of lake-terminating glaciers compared to land-terminating glaciers during 2017.*

Line 18 (Abstract): "exhibit greater acceleration". This refers to spatial, down-ice acceleration. For clarity, consider rephrasing to "exhibit greater rates of down-ice acceleration."

Amended.

Line 91 (Results): The text states median flowline velocities are 59 m yr^{-1} (lake) and 37 m yr^{-1} (land). Please clarify how these numbers were derived (e.g., "the median velocities calculated across all sampling boxes along the flowlines").

These values have been amended in keeping with the updated results, and the wording changed in text to:

Lines 95 – 98 (Results): *For each up-glacier distance, we calculated the median velocities for lake- and land-terminating glaciers. Median velocities were significantly higher at lake-terminating glaciers than land-terminating glaciers ($p < 0.05$), with overall median values of 43.14 m yr^{-1} and 36.21 m yr^{-1} , respectively.*

Line 93 (Results): "up-ice sample boxes". It is unclear which boxes this refers to; please specify a distance.

Amended:

Lines 98 – 99 (Results): *Whilst the two groups display insignificant differences at sample boxes between 5000 m and 9500 m up-ice...*

Line 94 (Results): The statistical notation ($p < 0.04$) should be ($p \leq 0.04$).

Amended.

Line 179 (Discussion): "median velocity is ~3.9 times higher". Based on lines 100–101, this would be exactly 4.0 times higher ($36 \text{ m yr}^{-1} / 9 \text{ m yr}^{-1}$). Please check or clarify.

Amended according to updated results.

Line 227 (Discussion): "marine glaciers" should be "marine-terminating glaciers."

Amended.

Line 275 (Methods): "an average of four 240 m² pixels". This appears to be an error. The text should refer to "four 240 m × 240 m pixels", or maybe the author want use other styles.

Thanks for spotting this- amended.

Reviewer #2 (Remarks to the Author):

Summary:

In this paper, the authors utilize a combination of two pre-existing datasets to contrast the velocity differences between glaciers with ice marginal lakes and those that are land-terminating. The study is built on an inventory of Greenland-Wide Ice Marginal Lakes produced from Sentinel 1 imagery in 2017 and then contrasted with a 2017 annual mosaic of ice surface velocities from the NASA MEaSUREs ITS_LIVE dataset. Statistical analyses between the two datasets reveal that glaciers with ice marginal lakes are almost 3 times as fast as their nearby land-terminating counterparts, and therefore greater attention should be paid to these types of glaciers in models of ice sheet evolution. The paper is very well-written, the figures are easy to understand, and the manuscript provides novel insight into lake-terminating glacier dynamics. I just have a few main comments which I include below.

Main Points:

1. My main piece of feedback is on the use of the terminology "ice marginal lake" vs. "proglacial lake" throughout your manuscript. The methods section specifies that you are only looking at glaciers that terminate in lakes, but the term "ice marginal" in your manuscript title and elsewhere in the text implies that you are also looking at glaciers that have ice-dammed lakes adjacent to them but may have no lake at the terminus. Since the focus of your paper is really on proglacial lake-terminus interactions, I suggest rephrasing throughout.

There is considerable ambiguity throughout the literature as to these two expressions; in previous publications, lakes that abut a glacier terminus are generally referred to as ice-marginal lakes, discriminating them from lakes at the front of, but not in contact with, a glacier terminus (often termed proglacial lakes). To maintain consistency with this use of the terminology, we have opted to use *ice-marginal lakes* here, but have amended the title to specify *proglacial ice-marginal lakes*, and changed the first reference to IMLs to the following:

Lines 37 – 40 (Introduction): *These changes have significantly reconfigured the ice sheet margin, particularly with a growth in the number and size of ice-marginal proglacial lakes (IMLs), which form as meltwater accumulates in the numerous topographic overdeepenings revealed during ice sheet retreat.*

2. I appreciate the authors use of pre-existing datasets to provide new insight into ice sheet evolution. While other manuscripts have contrasted glacier-average differences in velocities between land-terminating and lake-terminating glaciers, this manuscript dives deeper to examine whether these glaciers accelerate or decelerate near their termini. A main point of your paper that seems overshadowed to me is that only 32% of

lake terminating glaciers accelerate longitudinally towards the terminus, whereas prior studies cited (i.e., Holt et al. 2024) have emphasized the along-flow acceleration of glaciers with proglacial lakes. Your analysis of proglacial lakes of various sizes shows that there is more nuance to this than previously thought, and this point could be emphasized more throughout the paper. The cryosphere community's bias on studying the biggest, fastest retreating glaciers can often overlook nuanced findings like this that show glaciers can behave quite heterogeneously depending on their size, all of which are contributing to mass loss of the Greenland Ice Sheet.

Thank you for highlighting this. We agree that is an important point which could have been better acknowledged, and have added the following underlined text to the introduction and discussion:

Lines 85 – 87 (Introduction): *Overall, these findings demonstrate that GrIS outlet glaciers respond dynamically to IMLs, albeit variably, with implications for the inclusion of IMLs in ice sheet models.*

Lines 224 – 228 (Discussion): *Furthermore, we find that only 44% of glaciers exhibit a down-ice acceleration similar to that observed at Isortuarsuup Sermia. This discrepancy reflects large heterogeneity in the longitudinal flow regimes of lake-terminating glaciers across the ice sheet, wherein glaciers with velocities modified to the extent seen at Isortuarsuup Sermia are found to be amongst the most sensitive to lake forcing.*

3. Have you considered the impact of ice velocity seasonality on your results? As you mention in your discussion, the ice velocity of lake-terminating glaciers is influenced by several processes, such as subaqueous melt and subglacial hydrology, that all vary seasonally. How might using an annual velocity mosaic compared to the closest velocity data in time to the lake outline impact your findings?

We agree that lake-terminating glacier dynamics are influenced by processes which vary seasonally, such as subaqueous melt and hydrological inputs. In turn, we feel that if the study were to focus on shorter periods of analysis, the results would likely be biased towards periods of enhanced (or dampened) lake-ice interactions (i.e. enhanced summer velocities), which would make it difficult to accurately assess spatial trends, which is where we wanted the focus of the manuscript to lie. By using annually averaged velocity mosaics, our data represents normal lake glacier dynamics which can be robustly compared across years and between glaciers. Future research characterising the seasonality of velocity trends would certainly be of interest, but we suggest that this sort of analysis would constitute a full study.

4. While this study focuses on the pan-Greenland patterns in lake-terminating glacier ice velocity, I think it could be useful to include your results from individual glaciers in the supplementary material, while also pointing out specific examples in your results and discussion sections (see Slater et al. 2022 as an example: <https://www.nature.com/articles/s41561-022-01035-9>). This manuscript sets the stage for a harder look into glacier-lake interactions, so providing your dataset and specific examples will facilitate the future study of the dynamics that are driving the broadscale patterns you observe.

Thank you for this suggestion. We will make the full dataset available as a supplementary file, which includes velocities results for every sampled glacier alongside their ID numbers, which can be matched to the associated sampling box shapefiles. To

highlight examples of the most extreme behaviour in response to this comment, we have added a figure to the supplement (Fig. S3) which shows velocity profiles and locations of four glaciers; three which exhibit the greatest towards-terminus speed ups, and one which exhibits particularly fast along-flowline velocities.

5. I would recommend adding a section to the end of your methods on the statistical tests you performed. Which tests did you choose to examine your data, why, and what assumptions go into them?

We have added the following section to Methods (4.2. Ice Surface Velocities):

Lines 334 – 337 (Methods): *We use the Mann-Whitney U test to statistically assess the similarity of lake- and land-terminating glacier velocities and topographic characteristics, and lake-terminating glacier velocities across lake area bins. This test was chosen for its suitability for continuous data within two independent samples with skewed distributions.*

Line by line comments:

L32: “More than half of this mass loss occurs via outlet glaciers” – where else would this mass loss occur? Do you mean marine-terminating outlet glaciers in comparison to land-terminating? Or dynamic loss via calving vs. surface melting? Specify.

Amended:

Lines 34 – 35 (Introduction): *More than half of this mass loss occurs via ice sheet outlet glaciers,...*

L51: Should “ablation” be “frontal ablation”?

Amended.

L82: Can you be more specific about the “implications for the inclusion of IMLs in ice sheet models”. For those unfamiliar with how they are currently implemented or not, what specifically about your findings will be relevant for ice sheet models?

We have addressed this at the end of the discussion with the following text:

Lines 267 – 271 (Discussion): *Our findings carry implications for mass balance estimates derived from the input-output method, which typically account for ice flux only at the marine margins of the GrIS (Mankoff et al., 2021³⁴), but perhaps more importantly, they also suggest that present and future mass loss may be underestimated if glacier-lake interactions are omitted from model scenarios.*

L92: Should “negligible differences” be “insignificant differences”? Or how do you define negligible

Changed to “*insignificant*”.

L99: Should “at lake- and land-terminating” be “between lake- and land-terminating”?

Amended.

L103: Specify what IQR means upon its first use

Amended.

L106: Why are there such limited sample numbers outside of the SW, N, and NE regions? Does it have to do with the datasets and size criteria you used, or just the number of glaciers with ice marginal lakes in those regions? Could be helpful to specify.

This is a product partly of the criteria used to identify sample glaciers, and partly of the spatial distribution of lakes and lakes of sizes $> 1 \text{ km}^2$. The regions with smaller sample numbers ($n =$ four in CW, two in NW and three in SE) collectively hold $\sim 32\%$ of all lakes abutting the ice sheet margin, including those $< 1 \text{ km}^2$ (How et al., 2021), but crucially have smaller average lake areas (0.98 km^2 in CW, 0.45 km^2 in NW, and 0.39 km^2 in SE) than the SW, NO and NE. This is reflected in the sample populations used here, since fewer glaciers met the 1 km^2 lake area threshold. We have added a sentence to this effect:

Lines 285 – 286 (Methods): *Note that smaller sample numbers in CW, NW and SE reflect a lower average lake area ($< 1 \text{ km}^2$) in these regions.*

L120: Along the lines of my comments above, it could be useful to name these three glaciers that show the largest acceleration for subsequent studies.

To the best of our knowledge, these glaciers have not been previously inventoried or named in the literature, so we have amended the text to include their ID numbers, meaning that readers can find each glacier in the supplementary results and shapefiles.

Lines 131 – 133 (Results): *The three outlet glaciers with the most substantial longitudinal accelerations show velocity increases of 160% (ID 50), 151% (ID 41) and 133% (ID 32), respectively (Fig. S3).*

L129: It would be helpful to put a description of the error bars in Figure 1a into the Figure 1 caption

Amended:

Lines 155 – 156 (Results): *The IQR of velocity values across the sample population is shown by the box height, with capped bars representing the min and max values.*

L153: Should “occupy the largest lakes” be “enter the largest lakes”?

Amended.

L157: Add a label for the 25th percentile in area to Fig. 2 for easier interpretation

Thanks for this suggestion. Since the percentile would land so closely to the existing labels (which refer to glacier rank in terms of lake area, including rank #25), we feel that the figure would become unnecessarily cluttered, especially as the 25th percentile should be fairly evident given that there are 102 glaciers in the dataset.

L165: How did you choose the lake size categories for looking at the lake-area vs. down-ice velocity changes (Table S2)? If these size categories are shifted, how does this change your findings of significant vs. insignificant? I'd be curious to see a line plot of percentage change in velocity vs. lake size to discern how ice acceleration depends on lake size.

We chose these lake area bins to allow a comparison between the smallest and largest glaciers, whilst avoiding extensive variability between sample numbers within each bin. To demonstrate the relationship between down-ice change in velocity and lake area

more generally (i.e., outwith these bins), we have added a scatter plot showing the percentage change in glacier velocity between 200 m and 500 m up-ice at every glacier, plotted against the associated lake area (Fig. S4). This plot is referenced in the results section of the main text:

Lines 183 – 186 (Results): *Glaciers occupying the smallest lakes typically decelerate towards their termini, whilst a high number of glaciers which terminate in the medium-sized and largest lakes accelerate towards their termini (see also Fig. S4).*

L181: Add citation for the 5 km inland distance surpassing the grounding line location

We felt that this was a reasonable assumption to make, but since it isn't evidenced in the literature, we have removed it from the manuscript.

L194: Specify what Isotuarsuup Sermia is flowing 1.2-3.4x faster than

Amended:

Lines 217 – 219 (Discussion): *...Holt et al. (2024) observed flow velocities 1.2 to 3.4 times faster at lake-terminating Isortuarsuup Sermia than land-terminating Kangaasarsuup Sermia (2 km up-ice),...*

L196: Change “lacustrine glaciers” to “lake-terminating glaciers” to match previous wording

Amended.

L208-214: Are there any lakes in your dataset that you could look into for more of a case study on these claims? Somewhere you could create a time series of lake-size, overall glacier velocity longitudinal profile, and near terminus acceleration? I suspect that the relationship between velocity acceleration and lake size has more to do with a ratio of terminus thickness to lake depth, rather than lake age. For example, if a glacier terminates in a very large lake, but is just about to retreat onto land, I imagine it would have a much different velocity profile compared to a glacier terminating in the same size lake that still has substantial subaqueous contact. Either diving into a case study, or adding this caveat would help strengthen your claims.

Thank you for this suggestion. Since a temporal analysis such as this would require several datasets at a temporally fine resolution with high coherency, we have opted to add the following caveat to the discussion:

Lines 238 – 252 (Discussion): *However, the most significant longitudinal velocity accelerations occur into medium-sized lakes, implying that, once an IML reaches a certain age or depth, or indeed when a glacier begins to retreat out of a lake basin over a prograde slope, it perpetuates a towards-terminus velocity increase but no longer enhances this acceleration over time. This non-linear relationship likely reflects a complex set of glacier-lake interactions (Carrivick et al., 2020), which depend on temporally-evolving, glacier-specific factors such as terminus ice thickness, lake water depth, lake thermal regime, terminus morphology, and basal effective pressure (Mallalieu et al., 2020³⁰; Tsutaki et al., 2013¹⁹).*

Whilst our results reveal a widespread glacier response to IMLs, the mechanisms driving accelerated flow remain unclear. We expect that this behaviour can be partly attributed to buoyancy forces at the glacier front, governed by the relationship between ice thickness

and water depth, which reduce effective pressure and thereby lower basal drag (Benn et al., 2007). Enhanced velocities are also likely influenced by the debuitressing effect of calving, which is controlled by local factors such as ice-cliff geometry, lake water temperature, and ice buoyancy (Liu et al., 2020; Röhl et al., 2017; Mallalieu et al., 2021).

L249: Specify what dimensions you used to find a comparable land-terminating glacier. Length, width, height, ice flux? Also note that in Supp Note 1 you refer to section 2.1, which I think should be 4.1.

To address both this suggestion and a similar comment from Reviewer 1 (Comment 4), the revised manuscript includes new analysis of the similarity between lake- and land-terminating glacier populations in terms of their slope, elevation and aspect at each distance up-ice (Fig. S5). Unfortunately, quantifying dimensions/ catchment areas for each glacier is much trickier, since the sampled lake-terminating glaciers are not yet described in glacier inventories of catchment datasets. We have given a full explanation of the new analysis in our response to R1 Comment 4.

Supp. Note 1 amended.

L287: Recommend re-ordering the labelling of subfigures here so they go in clockwise order and for example, panels b and c are next to each other.

Amended.

-Nicole Abib

Reviewer #3 (Remarks to the Author):

This manuscript presents a comprehensive, pan-Greenland assessment of how ice-marginal lakes (IMLs) influence outlet glacier dynamics. The authors compared longitudinal velocity profiles of 102 lake-terminating glaciers with a matched set of land-terminating glaciers across all six regions of the Greenland Ice Sheet, using 2017 velocity data from NASA's ITS_LIVE dataset and lake inventories from How et al. (2021).

This is a well-written paper with a clear and logical structure. The research question is timely and relevant, the methodology is generally sound, and the results provide important new insights into glacier-lake interactions at the ice sheet scale. The manuscript would make a valuable contribution to Communications Earth & Environment. However, I have several suggestions for improvement before publication.

(1) In the method section 4.1, I would suggest to give a statistical characterization of lake-terminating and land-terminating glacier samples. Currently, the manuscript states that land-terminating glaciers were selected based on "similar dimensions," but without quantitative metrics, it's difficult to evaluate how well-matched these populations are. I recommend adding a table or figure in Section 4.1 that presents the distribution of key glaciological parameters for both lake-terminating and land-terminating glaciers, including: Glacier area, Elevation range (min, max, median), slope, Aspect/orientation, etc. This would allow readers to assess whether the two populations are truly comparable and whether any differences in velocity could be partially explained by systematic differences in glacier geometry or topographic setting.

Please see our earlier response to Reviewer 1 (Comment 4), which details a new set of analyses quantifying the topographic characteristics of the lake and land glacier populations. This analysis is now referenced in the methods section and has been added as a figure and a table to the supplementary information (Fig S5, Table S3):

Lines 289 – 295 (Methods): *To ensure that glacier characteristics other than terminus environment were consistent across the two sample populations, the average ice surface elevation, slope and aspect within each flowline box was extracted from ArcticDEM (10 m pixel resolution; Porter et al., 2022) and compared between lake- and land-terminating glaciers. We found no significant ($p < 0.05$) differences in the elevation, aspect or slope of lake- and land-terminating glaciers, except at 500 m up-ice, where glaciers flowing onto land are significantly steeper at the surface than those that flow into lakes ($p = 0.0003$) (Fig. S5).*

(2) In the method section 4.2, the manuscript states that flowlines were "systematically digitised to follow the line of fastest flow as observed in the ITS_LIVE 2017 velocity data, guided by surface features where visible." However, it is unclear whether this digitization was performed manually or through an automated algorithm. This distinction is critical for assessing reproducibility and potential bias in the analysis.

We have clarified this in text such that lines 310 – 313 now read:

Lines 310 – 313 (Methods): *To determine these locations, a flowline for every glacier was manually and systematically digitised to follow the line of fastest flow as observed in the ITS_LIVE 2017 velocity data, guided by surface features where visible in the MEaSUREs Greenland satellite image mosaics (Joughin, 2021³⁹).*

(3) In the method section 4.2, the choice of 500 m × 500 m sampling boxes spaced at 1500 m intervals appears reasonable but lacks justification. The scientific robustness of these parameters needs to be demonstrated, as different choices could potentially yield different results. I would recommend conducting a sensitivity analysis using a subset of representative glaciers (e.g., 5-10 glaciers spanning different sizes, velocities, and regions). Present the results as a supplementary figure or table demonstrating that the main conclusions are robust to these methodological choices. This would strengthen confidence in the findings and address potential reviewer concerns about parameter selection.

We have now carried out a sensitivity analysis investigating the influence of sampling strategy on the magnitude of sampled velocity values. This was performed using a selection six glaciers of varying location, terminus type and velocity. Box dimensions of 100 x 100 m, 200 x 200 m, 300 x 300 m, 400 x 400 m and 500 x 500 m were placed around centre points at 1500 m intervals along the flowline of each glacier. The 2017 velocity magnitude mosaic was then sampled as the mean velocity pixel value within each box at each flowline location.

Variability between the velocity values extracted from each box size was quantified by finding the mean percentage difference between box values, calculated at each flowline location for each glacier. The mean of these percentage differences is 5.70% (standard deviation 8.6%), suggesting that box size has a negligible effect on velocity results, and therefore that our conclusions hold true regardless of sampling box size.

This analysis is referenced in the Methods section and presented as a supplementary figure:

Lines 322 – 324 (Methods): *Results produced using this systematic sampling approach are robust to changes in box dimensions, when tested across glaciers of various regions and flow regime (Fig. S7).*

(4) The reference list does not conform to the formatting requirements of Communications Earth & Environment. The journal follows a numbered citation system with a specific format for references. Please carefully review the journal's author guidelines and reformat all references accordingly.

The reference list has been updated to conform exactly to the journal guidelines.

(5) Some abbreviations should be given in full name when they appear first time in the main text, e.g. IQR in Line 103; SW, CW, NW and NE in Line 122;

Amended.

(6) Add (c) in Line 137; N or NO should be consistent in Line 292; four 240 m² pixels, do you mean four 240m * 240m pixels?

Thanks for spotting this- amended.

Ice-marginal proglacial lakes enhance outlet glacier velocities across Greenland

Response to Reviewers – Version 2

Summary

We are very grateful to the three reviewers for their work on our manuscript and their positive and constructive comments. Please find our responses to comments from Reviewer #1 in blue below.

Reviewer #1 (Remarks to the Author):

The authors have addressed all comments raised in the first round of review. I am satisfied with the revised manuscript and have no further major comments. I have only a few minor suggestions for the authors to consider to further polish the discussion and clarify limitations:

1. Lake area vs. lake depth

The analysis in the manuscript establishes a relationship between glacier velocity and lake area (Figure 2). However, the physical mechanisms driving flow acceleration (buoyancy/flotation and thermal undercutting) are primarily functions of water depth rather than surface area. While I understand that bathymetry data is not available, it would be beneficial to explicitly acknowledge in the discussion that "lake area" is an imperfect (a compromise) proxy for depth. This would help explain the "complex" non-linear relationship observed, where medium-sized lakes also induce significant acceleration.

Thank you for this suggestion. We have added the following sentence to the discussion:

Line 212 – 213 (Discussion): "In this study, since bathymetry data is currently limited, we use lake surface area as an imperfect proxy for volume."

2. Bed topography controls

The revision clarifies that surface slope was used to match the control group glaciers. However, it is likely that many land-terminating glaciers reside on prograde beds, while lake-terminating glaciers lie on retrograde (overdeepened) beds. Since bed topography was not explicitly controlled for, acknowledging this as a potential contributing factor to the observed velocity differences (beyond just the "lake effect") would add rigor to the interpretation.

We agree that this is an important point, and have added it to the discussion:

Line 229 – 232 (Discussion): "Beyond lake effects, the observed velocity differences may also arise from contrasting bed topographies, with lake-terminating glaciers more likely to occupy retrograde beds and land-terminating glaciers typically lying on

This manuscript presents a pan-Greenland analysis of the influence of ice-marginal lakes (IMLs) on the dynamics of outlet glaciers. The authors compare the longitudinal velocity profiles of 102 lake-terminating glaciers with a corresponding set of 102 land-terminating glaciers using publicly available satellite data from 2017. The primary findings are that lake-terminating glaciers are significantly faster than their land-terminating counterparts (by ~295% at the terminus), with this effect extending approximately 5 km inland. Furthermore, about one-third of lake-terminating glaciers exhibit acceleration towards their terminus, a behaviour that is rare in land-terminating glaciers. The study concludes that IMLs exert a crucial control on Greenland Ice Sheet outlet glacier dynamics, a factor that should be incorporated into future ice sheet models.

General Comments

This study addresses a significant knowledge gap in our understanding of the role of IMLs in the Greenland Ice Sheet. The expansion of IMLs is a well-studied result of climate change, but their integrated, ice-sheet-wide impact on glacier dynamics has remained unquantified. The authors utilized a public lake inventory and ice velocity dataset, providing a large-scale quantification of this effect. The study is novel in its scope, and the findings have clear implications for the ice sheet modelling community.

However, the study in its current form has several limitations related to its temporal scope and methodological choices that raise significant questions about the robustness and context of the conclusions. While the key finding is likely correct, the issues detailed in the following sections must be addressed through a major revision before the manuscript can be considered for publication.

Major comments:

1. Lack of historical context and temporal analysis

The study's most significant limitation is its reliance on a spatial "snapshot" analysis. While the authors use 2015–2019 data to verify their 2017 results, this recent focus leaves a critical question unanswered: is the observed velocity difference a recent phenomenon, or has it been a long-standing feature of the ice sheet? The introduction states that IMLs “have the propensity to alter glacier dynamics and enhance rates of ice mass loss,” but it is unclear if the lake-terminating glaciers studied were already flowing faster than their land-terminating counterparts before any recent enhancement. Therefore, a temporal analysis is necessary to support the claim of "enhancement," which currently implies a temporal speed-up that the study does not show.

While I do not expect a full temporal analysis within the scope of a revision, it is reasonable to ask the authors to perform a preliminary analysis for an earlier period (e.g., a year or block of years before 2010) using the available ITS_LIVE velocity data. This would provide a first-order test of whether their conclusion holds true for a different climatic period. Additionally, the authors should clearly state that their study demonstrates an "enhanced velocity state" and a different spatial velocity profile in the presence of lakes but does not track glacier velocity changes through time. This discussion should frame the study as a crucial baseline that powerfully motivates future research into the temporal co-evolution of IMLs and glacier dynamics.

2. Use of an old version of the velocity dataset

The authors have used version 1 of the ITS_LIVE velocity dataset, which has a spatial resolution of 240 m. A newer, improved version 2 has been available for over a year and has a higher resolution of 120 m. Scientific best practice dictates using the most current and highest-quality data available. The research targets are land- and lake-terminating glaciers, which are relatively narrow compared to marine-terminating glaciers. Thus, the use of lower-resolution data may impact the study's precision, as a 240 m pixel size may be insufficient to accurately capture flow dynamics at the margins and terminus.

The authors should justify using the older dataset. Ideally, they should be required to re-run their entire analysis using the 120 m ITS_LIVE version 2 data. This would significantly increase confidence in the precision and robustness of their quantitative results.

3. Insufficient investigation of physical mechanisms

The manuscript establishes a pan-Greenland correlation between IMLs and higher glacier velocities but provides little insight into the underlying physical mechanisms. While the authors identify several potential drivers in the Introduction and Discussion (including calving, reduced buttressing, and the alteration of subglacial water pressures), the analysis does not test or discuss in any depth the relative importance of these processes, leaving the "how" and "why" behind their main finding largely unanswered.

I suggest the authors expand their Discussion to provide a more thorough synthesis of the underlying physics and the key local variables that control each process (e.g., the role of lake depth in calving; the influence of bed properties on subglacial hydrology).

4. Ambiguity in the selection of the control group

The methodology for selecting the land-terminating control glaciers is described qualitatively as choosing the "nearest land-terminating glacier with similar dimensions." This is a critical step in the methodology, and the lack of quantitative criteria could introduce potential bias (e.g., are lake-terminating glaciers systematically located in regions with steeper slopes or different subglacial conditions that would facilitate faster flow?).

The authors should strengthen the Methods section by providing a more detailed, and if possible, quantitative description of the criteria used for pairing glaciers (e.g., ranges of width, catchment area, slope, etc.). This would increase confidence that the primary differentiating variable between the two groups is indeed the terminus environment. I also suggest the authors share the related shapefiles with the reviewers, even at the peer-review stage, to allow for verification.

Minor Points / Line-by-Line Comments

Title: As noted, "Enhance" could be misinterpreted as a temporal speed-up. The authors should ensure the text clarifies that the study demonstrates an enhanced velocity state.

Citation Style: There appear to be two citation styles in the manuscript. Please check the journal requirements and use a consistent style.

Line 14 (Abstract): The value "295% faster" should be clarified. The reported median values (36 m yr⁻¹ and 9 m yr⁻¹) would suggest a 300% difference.

Line 18 (Abstract): "exhibit greater acceleration". This refers to spatial, down-ice acceleration. For clarity, consider rephrasing to "exhibit greater rates of down-ice acceleration."

Line 91 (Results): The text states median flowline velocities are 59 m yr⁻¹ (lake) and 37 m yr⁻¹ (land). Please clarify how these numbers were derived (e.g., "the median velocities calculated across all sampling boxes along the flowlines").

Line 93 (Results): "up-ice sample boxes". It is unclear which boxes this refers to; please specify a distance.

Line 94 (Results): The statistical notation ($p < 0.04$) should be ($p \leq 0.04$).

Line 179 (Discussion): "median velocity is ~3.9 times higher". Based on lines 100–101, this would be exactly 4.0 times higher (36 m yr⁻¹ / 9 m yr⁻¹). Please check or clarify.

Line 227 (Discussion): "marine glaciers" should be "marine-terminating glaciers."

Line 275 (Methods): "an average of four 240 m² pixels". This appears to be an error. The text should refer to "four 240 m × 240 m pixels", or maybe the author want use other styles.